# The effectiveness of physical activity in asthma management: An overview of systematic reviews

**Wenrui Liu**[1,2,3], **Zhenzhen Feng**[1,2,3*], **Shangyue Song**[1,2,3], **Siyuan Lei**[1,2,3]

**1** Department of Respiratory Diseases, the First Affiliated Hospital of Henan University of Chinese Medicine, Zhengzhou, China, **2** The First Clinical Medical School, Henan University of Chinese Medicine, Zhengzhou, China, **3** Collaborative Innovation Center for Chinese Medicine and Respiratory Diseases Co-constructed by Henan Province & Education Ministry of P.R. China/Henan Key Laboratory of Chinese Medicine for Respiratory Diseases, Henan University of Chinese Medicine, Zhengzhou, China

\* huxifzz@163.com

## Abstract

### Background

Physical activity (PA) has become a promising complementary non-pharmacological intervention to improve exercise capacity, cardiopulmonary fitness, and quality of life in individuals with asthma. This overview systematically consolidates existing evidence to assess the clinical viability of physical activity as a scalable supplementary therapy for asthma management.

### Methods

We searched 12 electronic databases to identify systematic reviews (SRs) from inception until February 12, 2025, concerning the efficacy of physical activity in asthma management. Literature was independently reviewed, data extracted and verified by two researchers. A third author was designated to mediate any disputes concerning screening. The methodological quality of SRs was assessed using the A Measurement Tool to Assess SRs 2 (AMSTAR 2) checklist, and the certainty of evidence for key outcomes was evaluated using the Grading of Recommendations Assessment, Development and Evaluation (GRADE) framework.

### Results

This study analyzed 34 SRs (published 2000–2024) that included quantitative synthesis involving 113–2280 participants from asthma populations: adults (9 SRs), children (8 SRs), and mixed adult-child cohorts (17 SRs), with disease severity varying from mild to severe. Ten SRs were assessed as moderate to high quality by AMSTAR 2, whereas the other SRs were classified as low or very low quality. We evaluated the quality of evidence for SRs utilizing the GRADE evidence quality assessment framework. Thirteen moderate-quality evidence, and 51 low or very low-quality evidence

**Funding:** This work was supported by Innovative Research Group Project of the National Natural Science Foundation of China, 82205313;The Second Batch of Discipline Construction Project of Chinese Medicine, the Characteristic Backbone Discipline of Henan Province, China, STG-ZYX03-202123; Joint Fund of Science and Technology Research and Development Program of Henan Province (Cultivation of dominant Disciplines), China, 232301420071.

**Competing interests:** The authors have declared that no competing interests exist.

support the improvement of PA on the outcomes of asthma quality of life, asthma control, lung function, exercise capacity, and respiratory muscle strength.

## Conclusion

Engagement in physical activity has been demonstrated to markedly enhance asthma-related outcomes. Specific interventions provide targeted advantages in various domains: aerobic exercise enhances AQLQ scores and lung function metrics, including FEV1, FVC, PEF; breathing exercises improve AQLQ, FVC, and $PI_{max}$; yoga correlates with enhancements in AQLQ, FEV1, and FVC; aquatic exercise results in increased FEV1; and inspiratory muscle training yields improvements in FEV1, FVC, PEF, and PImax. Nonetheless, there exists an imperative requirement for more stringent studies to fortify the existing evidence base. Furthermore, due to the significant individual variability among asthma patients, the creation of personalized exercise prescriptions is expected to produce enhanced clinical results.

## Introduction

Asthma is a heterogeneous disease marked by persistent airway inflammation, presenting with wheezing, dyspnea, chest constriction, cough, and additional symptoms [1]. It is acknowledged as one of the five most critical pulmonary diseases by global respiratory organizations, impacting around 350 million individuals globally and imposing considerable economic and social burdens [2,3]. Individuals with asthma frequently encounter compromised lung function and diminished exercise capacity, resulting in inadequate symptom management, recurrent exacerbations, and a decreased quality of life. Historically, asthma management has emphasized symptom control and the mitigation of exacerbation risk [4]. Pharmacological interventions constitute the foundation of asthma management. Although asthma can generally be managed effectively with medication, apprehensions remain about the long-term negative consequences of inhaled corticosteroids (ICS) and/or oral corticosteroids [5]. In addition, inadequate compliance with asthma medication and the occurrence of steroid-resistant asthma in certain individuals pose additional obstacles to disease management [6]. Despite the advent of biologic therapies enhancing treatment alternatives, ambiguities persist concerning their long-term safety and efficacy in clinical settings [7]. Therefore, recognizing scalable and economical complementary therapies is essential for enhancing clinical outcomes and patient-reported experiences.

Physical activity, a non-pharmacological intervention, has been widely used to improve quality of life, mitigate asthma symptoms, and enhance exercise capacity, cardiopulmonary fitness, and muscular strength [8–20]. The World Health Organization characterizes physical activity as "any bodily movement generated by skeletal muscles that necessitates energy expenditure exceeding resting levels." [21].

Numerous SRs have investigated the effects of physical activity on asthma. However, their results are not wholly consistent and have not been thoroughly synthesized [8–17,20]. Consequently, a synthesis derived from published SRs is crucial to mitigate the shortcomings of individual studies and offer a more thorough comprehension of the impacts of physical activity on asthma.

## Materials and methods

### Protocol and registration

The Cochrane Handbook for SRs of Interventions states that the overview of SRs was conducted and documented in accordance with the Preferred Reporting Items for SRs and Meta-Analysis (PRISMA) 2020 statement [22]. The protocol (CRD42024520761) has been documented in the International Prospective Register of SRs (PROSPERO) (https://www.crd.york.ac.uk/PROSPERO/).

### Search strategy

Literature searches were conducted using PubMed, Embase, the Cochrane Library, Web of Science, PEDro, CINAHL, Scopus, Sportdiscus, Chinese National Knowledge Infrastructure, Wan Fang database, Chinese biomedical literature service system, and Chongqing VIP, up to February 12, 2025. No restrictions were placed on the origin of the SRs, the date of publication, or the language of publication. The search strategy included relevant words related to asthma and physical activity (including 'physical activity', 'exercise training', 'exercise therapy', 'endurance training', 'resistance training', 'muscle stretching exercises', 'upper limb training', 'Running', 'Jogging', 'Swimming', 'Walking', 'Qigong', 'Yoga', 'Pilates' 'Gymnastics', 'Neuromuscular Electrical Stimulation', 'Physiotherapy') and Systematic Review and was adjusted for each database. Comprehensive search methodologies are outlined in Appendix S1.

### Study selection

In EndNote X9, duplicate records were automatically eliminated following the importation of electronic database searches. The eligibility screening was performed independently by two researchers, Wenrui Liu and Shangyue Song. A third author, Zhenzhen Feng, was designated to resolve any disputes concerning screening.

   The inclusion criteria encompassed all the subsequent aspects: (1) Participants diagnosed with asthma, irrespective of age, sex, or severity; (2) the intervention encompassed any form of physical activity(defined as any bodily movement generated by skeletal muscles that necessitates energy expenditure exceeding resting levels); (3) studies compared physical activity against placebo, blank control, usual care, conventional drug therapy, or other non-pharmacological therapies; (4) at least one outcome was measured among quality of life (AQLQ), asthma control (ACQ), forced expiratory volume in one second (FEV1), forced vital capacity (FVC), peak expiratory flow (PEF), six-minute walk distance (6MWD), maximal oxygen consumption (VO2max), and maximal inspiratory pressure (PImax); (5) The SRs were founded on randomized controlled trials concerning physical activity for asthma.

   The exclusion criteria included any of the following:(1) Participants with additional pulmonary disease complications aside from asthma; (2) lack of access to the complete text; and (3) duplicate publications of the same SRs or updates of the same SRs will retain only one instance.

### Data extraction

Wenrui Liu and Shangyue Song independently extracted the review characteristics (including first author, article title, publication year, sample size, intervention measures, etc.) using Microsoft Excel software, and subsequently cross-verified the data for accuracy. Disputes were settled through team deliberation or arbitration by the third reviewer, Zhenzhen Feng.

## Assessment of included SRs

Wenrui Liu and Shangyue Song independently evaluated the quality of the included reviews. Disagreements were settled through team deliberation or arbitration by the third reviewer, Zhenzhen Feng. The AMSTAR-2 and GRADE tools were utilized to evaluate methodological quality and the certainty of evidence in the systematic reviews included. However, these instruments were employed exclusively for assessment purposes and did not affect the criteria for study selection.

## Methodological quality

AMSTAR 2 [23] was used to assess the methodological quality. The quality of each SR was categorized into four tiers (high, moderate, low, and critically low) according to the quantity of critical deficiencies (indicated by a 'no' rating in items 2, 4, 7, 9, 11, 13, and 15) or non-critical shortcomings (indicated by a 'no' rating in other items). 0–1 non-critical item non-conformities is considered to be of high quality, > 1 non-critical item non-conformance is considered medium quality, 1 critical item non-conformance (with or without non-critical item non-conformity) is considered low quality, and > 1 critical item non-conformance is considered to be of critical low quality, regardless of non-critical item non-conformity.

## Evidence quality

The GRADE system was used to evaluate the quality of evidence. RCTs were first set as high quality evidence, and then the outcomes were downgraded by considering five aspects: limitations, inconsistency, indirectness, imprecision, and publication bias. if there is no downgrading, high-quality evidence; One was downgraded to moderate, two to low, and three or more to very low.

## Strategy for data synthesis

Due to considerable heterogeneity in the fundamental characteristics of the original studies incorporated in the SRs (e.g., exercise modality, intervention frequency, duration, and study population), a descriptive analysis was performed, categorized by exercise type.

# Results

## Literature search

A total of 3,465 potentially pertinent studies were identified. Subsequent to the elimination of 1,362 duplicate articles, 1,998 papers were excluded following title and abstract screening. Moreover, 79 studies that failed to satisfy the inclusion criteria were excluded after a comprehensive review of 113 articles. A total of 34 SRs were incorporated into the analysis. Fig 1 shows the literature retrieval and screening process, while Appendix S2 contains a list of excluded studies along with the primary reasons for their exclusion.

## Characteristics of the SRs

The 34 included SRs, published between 2000 and 2024, synthesized data from 5 to 30 primary studies, encompassing 113–2,280 participants. The interventions analyzed comprised aerobic exercise (4 SRs), breathing exercise (9 SRs), respiratory muscle training (3 SRs), yoga (2 SRs), and aquatic exercises (1 SRs). The research included asthma patients from diverse age categories: adults (9 SRs), children (8 SRs), and mixed adult-child cohorts (17 SRs), with disease severity varying from mild to severe. Table 1 presents a detailed summary of the study characteristics.

## Methodological quality with AMSTAR 2

The methodological quality of the included SRs was evaluated using the AMSTAR 2 tool. Among the 34 SRs, 24 (70.6%) were classified as low or critically low quality, whereas 10 (29.4%) exhibited moderate to high methodological rigor. The

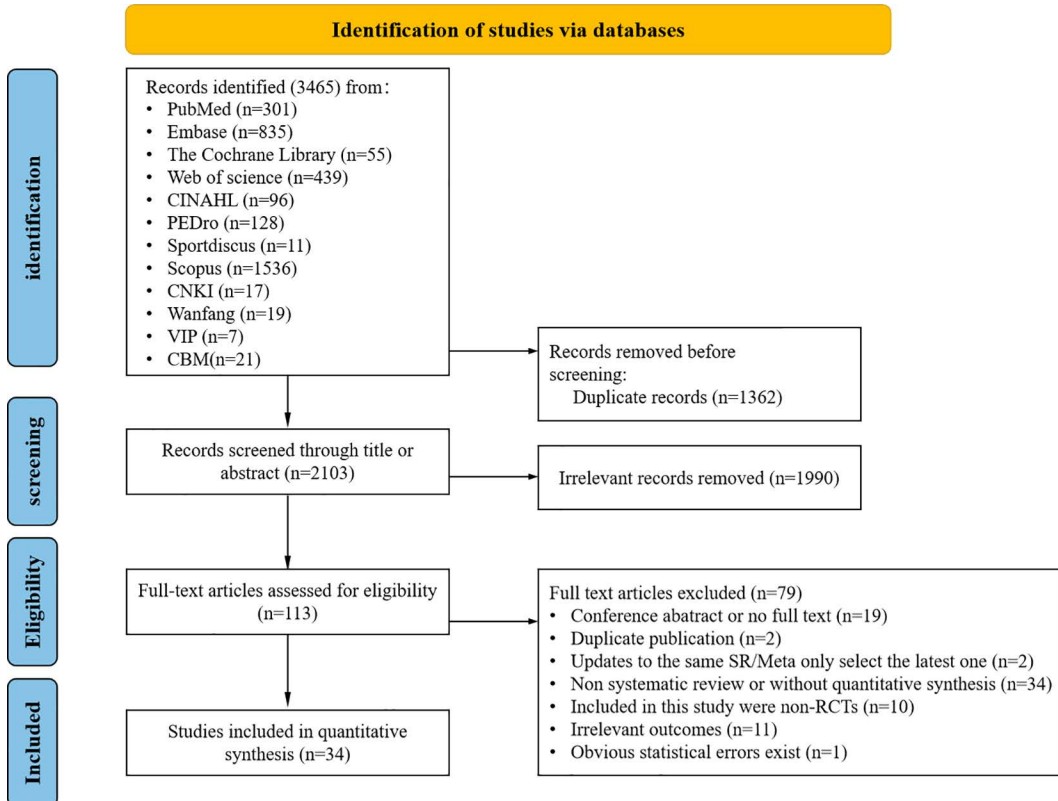

**Fig 1. Flowchart of literature selection.**

principal limitations noted were the lack of a registered protocol in 50.0% of SRs, the failure to report a list of excluded studies in 21 SRs (61.7%), and the omission of funding source information for primary studies in 30 SRs (88.2%). Of the reviewed SRs, 32 (94.1%) offered thorough descriptions of the included studies, 30 (88.2%) employed suitable methodologies to evaluate the risk of bias in individual studies, and 28 (82.3%) performed exhaustive literature searches. Moreover, all included SRs (100%) used suitable statistical techniques for data synthesis, 32 SRs (94.1%) accounted for heterogeneity in their analyses, and 22 SRs (64.7%) disclosed conflicts of interest and funding sources. The AMSTAR 2 scoring outcomes are shown in Fig 2.

## Evidence quality with GRADE

The results of the included SRs were synthesized through a descriptive analysis method, and the GRADE system was utilized to evaluate the quality of evidence for each outcome. Appendix S3 contains evaluation procedures.

## Quality of life (AQLQ)

Among 17 SRs reported AQLQ, 13 SRs (5 mixed, 4 adults, 4 children) demonstrated statistically significant enhancements in AQLQ scores (moderate: 4; low: 6, very low: 3). All 3 AE [14,29,40] SRs (2 mixed, 1 children) reported significant improvement (low: 3); both Yoga SRs [31,35] (1 mixed, 1 adults) indicated significant enhancements (low: 1; very low: 1); all 3 BE SRs [9,26,32] (1 mixed, 2 adults) revealed significant improvement (moderate: 2; very low: 1); and the single IMT SR [36] (mixed) found no significant improvement (very low). Details are shown in Fig 3.

**Table 1. Characteristics of the Included SR.**

| Reference | Trials (Sample Size) | Ages | Severity of Asthma | Intervention Treatment Group | | | | | Control Group | Outcomes |
|---|---|---|---|---|---|---|---|---|---|---|
| | | | | Type | Duration | Time | Frequency (Times/Week) | Intensity | | |
| Ram, 2003 [24] | 5(180) | Adults | Mild to severe | IMT | 3weeks~6months | 30min | 5~6 | 15~80% PImax | Sham or no inspiratory training | FEV1, FVC, PImax, |
| Holloway,2009 [25] | 7(403) | Mixed adult and children | Mainly mild to moderate | BE | 1~16weeks | 20~45 min | 2~7 | – | Education or inactive control | FEV1, FVC |
| Ram,2005 [19] | 13(455) | Mixed adult and children | – | PA | ≥4weeks | ≥20 min | 2~3 | – | Non-PA | FEV1, FVC, VO2max |
| Burges,2011 [9] | 30(1767) | Mixed adult and children | Mild to severe | BE | 1week~6months | 45min~2.5h | 1~5 | – | Non-BE or blank control | AQLQ, PEF, FEV1 |
| Eichenberger, 2013 [10] | 15(2059) | Mixed adult and children | Mild to severe | PA | ≥1weeks | ≥5 training session | ≥2 | – | Non-PA | FEV1, VO2max, PEF |
| Silva,2013 [11] | 5(113) | Adults | Mild to severe | IMT | 3~25weeks | 10~30 min | 6~12 | 15~60% of resistance | Sham or no inspiratory training device | PImax, FEV1, FVC |
| Carson,2014 [12] | 21(772) | Mixed adult and children | Any degree | PA | ≥4weeks | ≥20 min | ≥2 | – | Non-PA | VO2max |
| Cramer,2014 [13] | 14(824) | Mixed adult and children | – | Yoga | 2weeks~54months | 15min~4h | 1~5 | – | Usual care or pharmacotherapy or no intervention | FEV1, FVC |
| Santino,2020 [26] | 22(2280) | Adults | Mild to moderate | BE | 2weeks~6months | 10min~4h | 1~7 | – | Inactive control | AQLQ, ACQ, FEV1 |
| Wu,2020 [14] | 22(874) | Mixed adult and children | Any degree | AE | ≥4weeks | ≥20 min | ≥2 | – | No intervention or receiving the same level of education or medication | FEV1, FVC, PEF,AQLQ |
| Feng,2021 [15] | 9(418) | Adults | – | PA | 8~12weeks | – | – | – | Non-PA | AQLQ, ACQ, FEV1, FVC, PEF, 6MWD, VO2max |
| Chen,2022 [20] | 6(155) | Mixed adult and children | – | IMT | 6week~6months | 30 breaths or 30 min | 6-14 | – | Placebo | FEV1, FVC, PImax |
| Lista,2022 [16] | 11(270) | Mixed adult and children | Any degree | IMT | 6week~6months | 10~30 min | 3~7 | 15~80% PImax | Sham or simulated IMT or usual care or no intervention | PImax, FEV1, FVC, PEF, 6MWD |

*(Continued)*

| Reference | Trials (Sample Size) | Ages | Severity of Asthma | Intervention Treatment Group | | | | | Control Group | Outcomes |
|---|---|---|---|---|---|---|---|---|---|---|
| | | | | Type | Duration | Time | Frequency (Times/Week) | Intensity | | |
| Li, 2022 [17] | 14(1009) | Mixed adult and children | Stable Stage | PA, AE | 6~24week | — | — | — | Routine drug therapy | FEV1, FVC, PEF |
| Osadnik,2022 [18] | 10(894) | Adults | — | PA | ≥4week or 8 sessions | — | — | — | Usual care | ACQ, 6MWD, AQLQ |
| Shi,2022 [27] | 18(1073) | Adults | Mild to moderate | PA | ≥8weeks | — | ≥2 | — | Non-PA | AQLQ, FEV1, FVC, VO2max |
| Wang,2022 [28] | 13(598) | Mixed adult and children | Any degree | IMT | 6week~6months | 20~60 min | ≥2 | — | Control intervention | FVC, PImax |
| Zhu,2022 [29] | 18(1021) | Mixed adult and children | Any degree | PA, AE | 6week~4years | 8~90 min | 1~5 | — | Without exercise intervention | AQLQ |
| Franciele,2024 [30] | 10(393) | Mixed adult and children | mild to severe | Aquatic exercise | 5~24weeks | — | 1~6 | — | Control | FEV1, FVC |
| Anshu,2023 [31] | 15(1207) | Adults | mild to moderate | Yoga | — | — | — | — | Standard care, no treatment group, or usual care. | FEV1, FVC, ACQ, AQLQ |
| Freitas,2013 [32] | 13(906) | Adults | mild to moderate | BE | 2~24weeks | 15min~4h | ≥2 | — | Asthma education or with no active control group | AQLQ |
| McLoughlin,2022 [33] | 4(176) | Adults | severe | PA | ≥2weeks | 30~60 | 2~5 | moderate-vigorous | Usual care or sham intervention | ACQ, AQLQ |
| Ram,2000 [34] | 8(226) | Mixed adult and children | any degree | PA | ≥4weeks | ≥ 20 min | ≥2 | — | Control intervention | VO2$_{MAX}$ |
| Yang,2016 [35] | 15(1048) | Mixed adult and children | mild to moderate | Yoga | 2weeks~54months | 15min~4h | ≥2 | — | Usual care or sham intervention or no intervention | AQLQ, FEV1, FVC |
| Yi,2021 [36] | 10(456) | Mixed adult and children | — | IMT | ≥1months | ≥ 20 min or 9 sessions | ≥2 | — | Usual care or sham intervention or no intervention | FEV1, FVC, PEF, 6MWD, AQLQ |
| You,2024 [37] | 7(229) | Mixed adult and children | — | IMT | 3~12week | ≥ 20 min or 3 sessions | 2~10 | 15~60%PImax | Sham intervention or education | PEF, PImax |
| Jing,2023 [38] | 9(496) | Children | mild to moderate or stable stage | PA | 6~24week | 30~60 min | 1~3 | — | Non-PA | FEV1, FVC, FEF, PAQLQ |
| Liu,2021 [39] | 22(1346) | Children | nonacute | PA | 4~24week | 20~60 min | 1~14 | — | Routine activities or Routine drug therapy | 6MWT, PAQLQ |

(Continued)

**Table 1.** (Continued)

| Reference | Trials (Sample Size) | Ages | Severity of Asthma | Intervention Treatment Group | | | | | Control Group | Outcomes |
|---|---|---|---|---|---|---|---|---|---|---|
| | | | | Type | Duration | Time | Frequency (Times/Week) | Intensity | | |
| Ma,2024 [40] | 20(899) | Children | – | AE | 2d~12week | 20~60 min | – | – | Routine drug therapy or sham intervention or no intervention | FVC, PEF, FEV1, AQLQ |
| Xiang,2024 [41] | 6(333) | Children | Any degree | IMT | 5~12week | 20~30 min | 2~7 | 30~40%MIP | Sham inspiratory muscle training or usual care, | $PI_{MAX}$, FVC, FEV1, PEF |
| Yin,2019 [42] | 9(851) | Children | Any degree | PA | – | – | – | – | Non-PA | FEV1, FVC, PEF |
| Zhou, 2023 [43] | 15(802) | Children | – | PA | 5~16week | 20~60 min | 1~4 | – | Non-PA | FEV1, FVC, PEF |
| Liu Fang,2021 [44] | 14(900) | Children | nonacute | PA | 6~24week | AE 20~60 min, anaerobic exercise 30s | 3~7 | – | Usual care | AQLQ, 6MWD |
| Yang,2022 [45] | 15(936) | Children | mild to moderate | PA | 6~48week | 20~60 min | 2~5 | – | Usual drug therapy or without exercise training | VO2max, AQLQ |

**Abbreviations:** ACQ, asthma control questionnaire; AE, Aerobic exercise; AQLQ, asthma quality of life questionnaire; BE, Breathing exercises; PA, physical activity; IMT, Inspiratory Muscle Training; FEV1, forced expiratory volume in one second; FVC, Forced vital capacity; PEF, peak expiratory flow; PImax, maximal inspiratory pressure; $VO_{2max}$, maximal oxygen uptake; 6 MWD 6-min walk distance

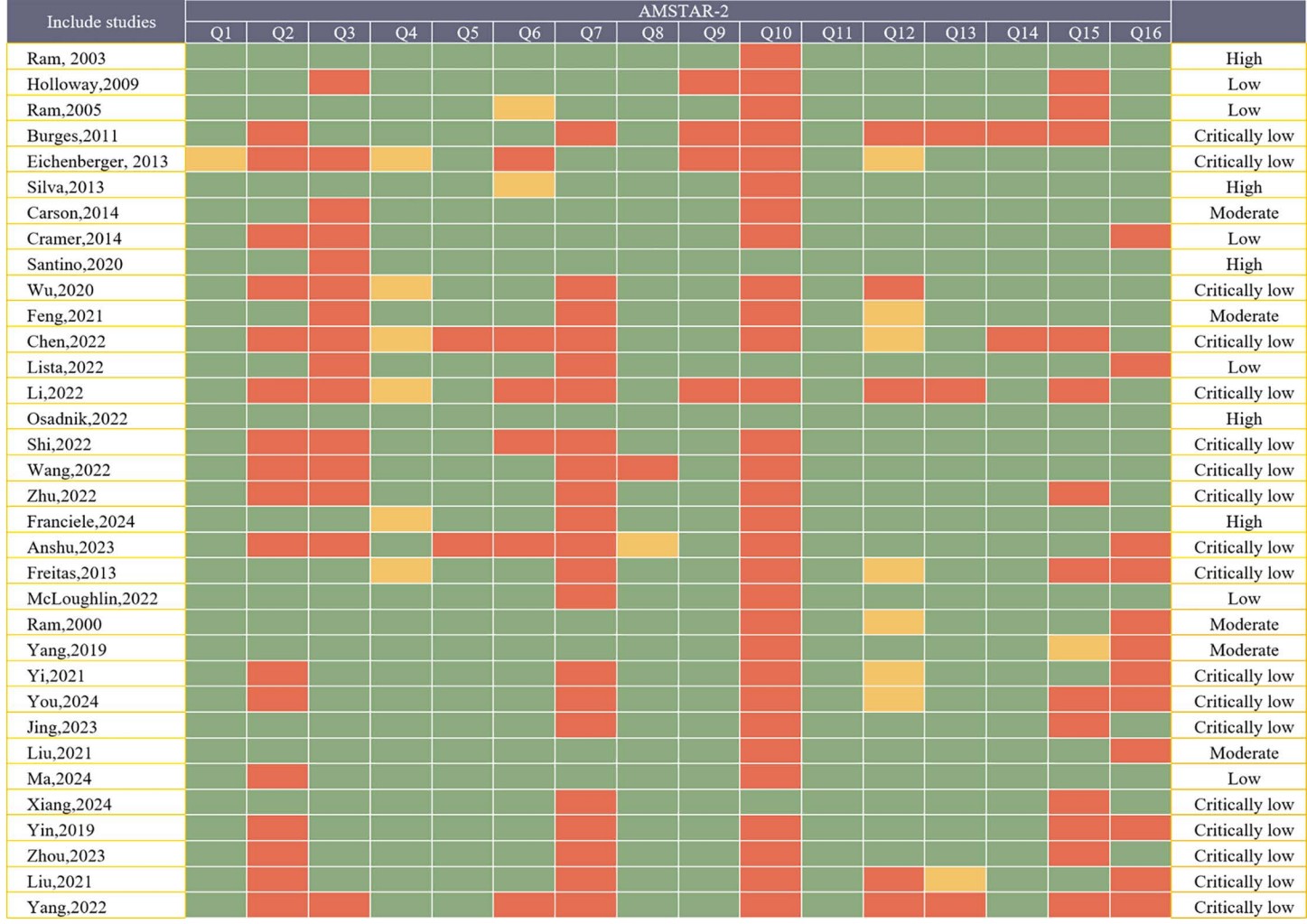

**Fig 2. Quality assessment for included systematic reviews using the AMSTAR 2 tool.** Q1. Question and inclusion included PICO; Q2. Protocol; Q3. Study design; Q4. Comprehensive search; Q5. Study selection; Q6. Data extraction; Q7. Excluded studied justification; Q8. Included studied details; Q9. Risk of Bias; Q10. Source of funding of included studies; Q11. Appropriate statistical methods for analysis; Q12 Rob on meta-analyses; Q13. Rob on individual studies; Q14. Explanation for heterogeneity; Q15-Publication bias; Q16. Conflict of interest. Green, Yes; Yellow, Partial Yes: Red, No.

## Asthma control (ACQ, Asthma symptom free days)

Among 5 SRs evaluated outcomes of the ACQ, 2 SRs(adults) showed statistically significant ACQ reduction (moderate:1, low:1). One Yoga SR [31] (adults) reported a non-significant increase in ACQ (very low); and 1 BE SR [26] (adults) found no significant ACQ reduction (low). Additional information is presented in Fig 4A.

Three SRs assessed days free of asthma symptom. Among these, one SRs [10] (mixed) indicated a significant improvement (very low). Fig 4B illustrates these findings.

## Lung function (FEV1, FVC, PEF)

Among 19 SRs reported FEV1 outcomes, 8 SRs (4 mixed, 1 adults, 3 children) demonstrated statistically significant improvement (moderate: 4; low: 2, very low: 2). Of 3 AE SRs [14,17,40], 2 SRs [14, 17] (2 mixed) reported improvement

                                                                

| Intervention | Ages | GRADE | Author | N | n | Quality | | MD/SMD (95% CI) |
|---|---|---|---|---|---|---|---|---|
| PA | M | moderate | Zhu,2022 | 9 | 534 | Critically low | | 0.41 (0.32, 0.51)* |
| PA | A | Very Low | Osadnik, 2022 | 2 | 442 | High | | 0.87 (-0.13, 1.86) |
| PA | A | Moderate | Feng, 2021 | 4 | 198 | Moderate | | 0.39 (0.02, 0.76)* |
| PA | A | Low | Shi, 2022 | 7 | 590 | Critically low | | -0.03 (-0.61, 0.54) |
| PA | A | Low | McLoughlin,2022 | 2 | 142 | Low | | -0.65 (-0.95, -0.35)* |
| PA | C | Low | Liu Fang,2021 | 10 | 794 | Critically low | | 1.28 (0.60, 1.95)* |
| PA | C | Low | Liu, 2021 | 10 | 714 | Moderate | | 1.28 (0.60, 1.95)* |
| PA | C | Very Low | Yang,2022 | 4 | 149 | Moderate | | 0.67 (0.43, 0.91)* |
| PA | C | Very Low | Jing,2023 | 5 | 298 | Critically low | | 1.38 (0.26, 2.50)* |
| AE | M | Low | Wu, 2020 | 5 | 233 | Critically low | | 0.81 (0.32, 1.30)* |
| AE | M | Low | Zhu,2022 | 9 | 192 | Critically low | | 0.52 (0.37, 0.67)* |
| AE | C | Low | Ma,2024 | 12 | 647 | Low | | 0.70 (0.14, 1.26)* |
| Yoga | M | Low | Yang,2016 | 5 | 375 | Moderate | | 0.57 (0.37, 0.77)* |
| Yoga | A | Very Low | Anshu,2023 | 5 | 645 | Critically low | | 0.26 (0.18, 0.34)* |
| BE | M | Moderate | Burgess, 2011 | 5 | 447 | Critically low | | 0.35 (0.00, 0.70)* |
| BE | A | Very Low | Freitas,2013 | 2 | 172 | Critically low | | 0.79 (0.50, 1.08)* |
| BE | A | Moderate | Santino, 2020 | 4 | 974 | High | | 0.42 (0.17, 0.68)* |
| IMT | M | Very Low | Yi,2021 | 2 | 70 | Critically low | | 1.21 (-2.11, 4.52) |

MD/SMD(95% CI)

**Fig 3. The forest plot of AQLQ.** Intervention indicates intervention type. Ages: M = mixed adult and child populations; A = adult populations; C = child populations. N = number of included primary studies; n = sample size of included studies; Quality = methodological quality assessed by AMSTAR 2; *= P ≤ 0.05 The same abbreviations apply Fig 3 to Fig 7.

a

| Intervention | Ages | GRADE | Author | N | n | Quality | | MD/SMD (95% CI) |
|---|---|---|---|---|---|---|---|---|
| PA | A | Moderate | Feng, 2021 | 5 | 215 | Moderate | | -0.25 (-0.51, 0.02)* |
| PA | A | Low | McLoughlin,2022 | 3 | 88 | Low | | 0.56 (0.10, 1.01)* |
| PA | A | Low | Osadnik, 2022 | 2 | 93 | High | | -0.46 (-0.76, -0.17)* |
| Yoga | A | Very Low | Anshu,2023 | 3 | 166 | Critically low | | 0.16 (0.15, 0.48) |
| BE | A | Low | Santino, 2020 | 1 | 115 | High | | -0.21 (-2.92, 2.50) |

MD/SMD(95% CI)

b

| Intervention | Ages | GRADE | Author | N | n | Quality | | MD/SMD (95% CI) |
|---|---|---|---|---|---|---|---|---|
| PA | M | Very Low | Eichenberger, 2013 | 2 | 109 | Critically low | | 8.90 (8.18, 9.61)* |
| PA | A | Low | Feng, 2021 | 2 | 94 | Moderate | | 3.35 (-0.21, 6.90) |
| PA | A | Very Low | Shi, 2022 | 4 | 216 | Critically low | | 0.93 (-0.39, 2.25) |

MD/SMD(95% CI)

**Fig 4. A. The forest plot of ACQ.** B. The forest plot of asthma symptom-free days.

(moderate: 1; very low: 1); 3 Yoga SRs [13, 31, 35], 1 SR [31] (1 adults)showed significant improvement (moderate); 3 BE SRs [9,25,26] (2 mixed, 1 adults) found no significant improvement(moderate 2, low 1); 2 IMT SRs [36,41] (1 mixed, 1 children) demonstrated significant improvement (moderate: 1; low: 1); and 1 aquatic exercise SR [30] (mixed) reported improvement (low). Fig 5A shows the details.

Twenty-one SRs evaluated FVC. Among these, 11 SRs (6 mixed, 2 adults, 3 children) showed significant improvement (moderate: 1, low: 7, very low: 4). Of 3 AE SRs [14,17,40] (2 mixed, 1 children), 2 SRs [14,17] reported benefits (low: 1; very low: 1); of 3 Yoga SRs [13,31,35], 2 SRs (1 mixed, 1 adult) [31,35] demonstrated improvement (low: 2); 1 BE SRs

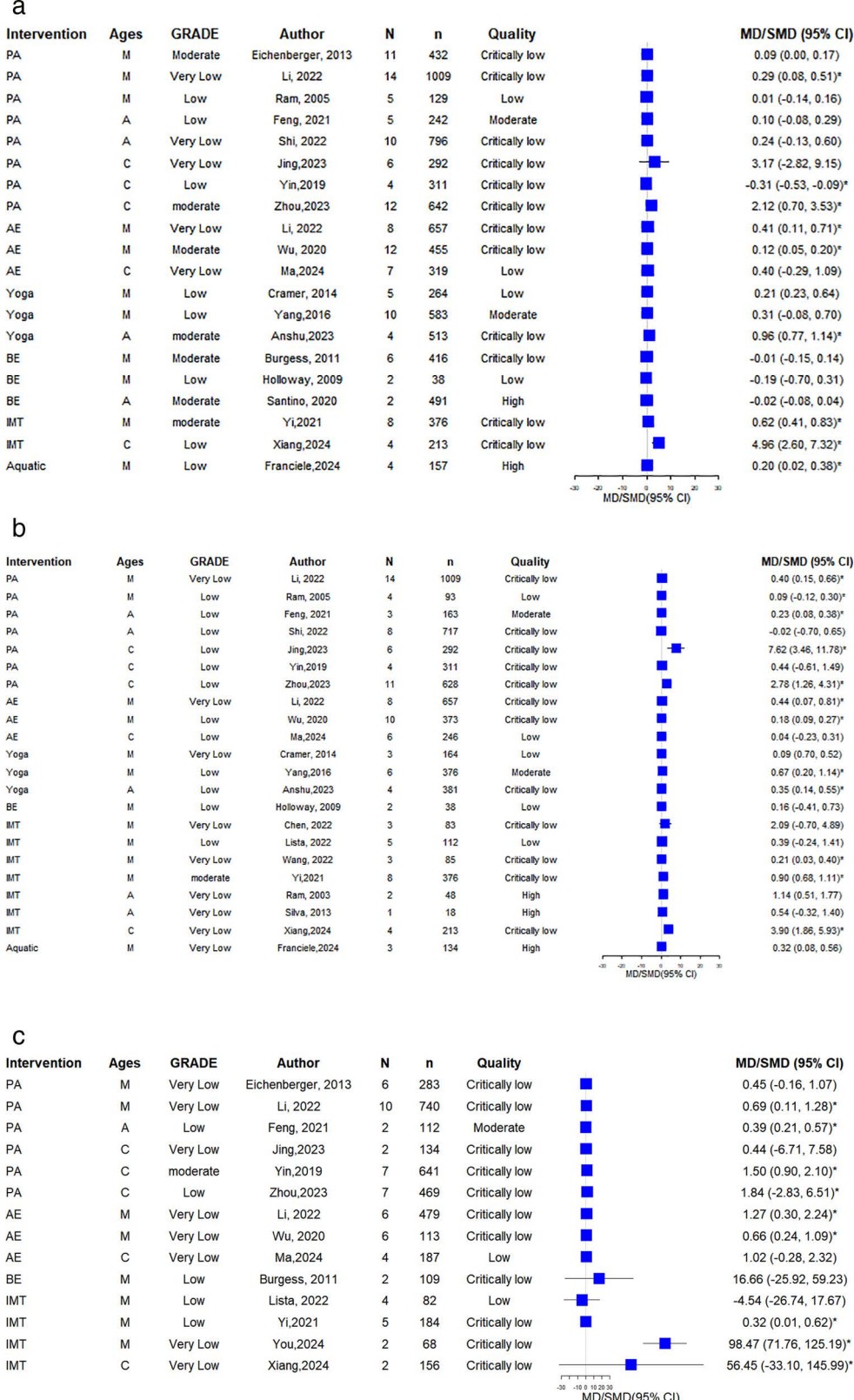

**Fig 5. A. The forest plot of FEV1.** B. The forest plot of FVC. C. The forest plot of PEF.

[25] (mixed) found no improvement; among 7 IMT SRs [8,11,16,20,28,36,41], 3 SRs [28,36,41] (2 mixed, 1 child) showed significant improvement (moderat:1, very low: 2); and 1 aquatic exercise SR [30] (mixed) found no FVC benefit (very low). Details are shown in Fig 5B.

Among 14 SRs assessed PEF, 9 SRs (5 mixed, 1 adult, 3 children), reported significant improvement (moderate: 1; low: 3; very low: 5). Of 3 AE SRs [14,17,40], 2 SRs [14,17] (2 mixed) showed benefits (very low: 2); 1 BE [9]SRs (mixed) found no improvement (low); and of 4 IMT [16,36,37,41] SRs, 3 SRs (2 mixed, 1 child) demonstrated improvement (low: 1; very low: 2). Information is presented in Fig 5C.

### Exercise performance (6MWD, VO2max)

Among 6 SRs evaluated 6MWD, 5 SRs (1 mixed, 2 adults, 2 children) reported significant increase (moderate: 1, low: 2, very low: 2). One IMT SR [36] (mixed) demonstrated significant increase (very low: 1). Fig 6A illustrates the details.

All 7 SRs (4 mixed, 2 adults, 1 children) assessed VO2max showed significant increases (moderate: 1; low: 5; very low: 1). Details are shown in Fig 6B.

### Respiratory muscle function (PImax)

Seven SRs evaluated PImax [8,11,16,20,28,37,41] (4 mixed, 2 adults, 1 child) demonstrated significant improvement (low: 4; very low: 3). Details are shown in Fig 7.

## Discussion

### Main findings

This study examined 34 SRs published from 2000 to 2024, encompassing 113–2,280 participants with asthma across diverse age demographics: adults (9 SRs), children (8 SRs), and mixed adult-child populations (17 SRs), with disease severity varying from mild to severe. The results indicate that physical activity correlates with significant enhancements in asthma-related metrics, encompassing the AQLQ, ACQ, FEV1, FVC, PEF, 6MWD, and VO$_2$max. Different exercise

a

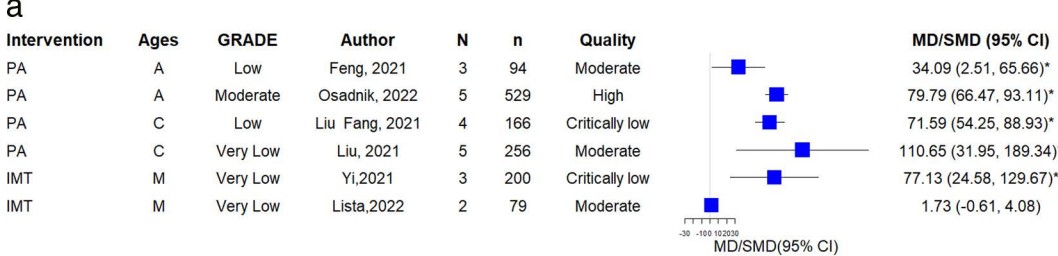

| Intervention | Ages | GRADE | Author | N | n | Quality | MD/SMD (95% CI) |
|---|---|---|---|---|---|---|---|
| PA | A | Low | Feng, 2021 | 3 | 94 | Moderate | 34.09 (2.51, 65.66)* |
| PA | A | Moderate | Osadnik, 2022 | 5 | 529 | High | 79.79 (66.47, 93.11)* |
| PA | C | Low | Liu Fang, 2021 | 4 | 166 | Critically low | 71.59 (54.25, 88.93)* |
| PA | C | Very Low | Liu, 2021 | 5 | 256 | Moderate | 110.65 (31.95, 189.34)* |
| IMT | M | Very Low | Yi,2021 | 3 | 200 | Critically low | 77.13 (24.58, 129.67)* |
| IMT | M | Very Low | Lista,2022 | 2 | 79 | Moderate | 1.73 (-0.61, 4.08) |

b

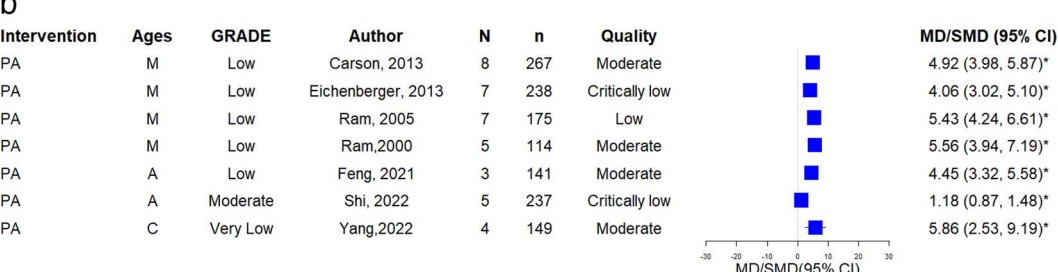

| Intervention | Ages | GRADE | Author | N | n | Quality | MD/SMD (95% CI) |
|---|---|---|---|---|---|---|---|
| PA | M | Low | Carson, 2013 | 8 | 267 | Moderate | 4.92 (3.98, 5.87)* |
| PA | M | Low | Eichenberger, 2013 | 7 | 238 | Critically low | 4.06 (3.02, 5.10)* |
| PA | M | Low | Ram, 2005 | 7 | 175 | Low | 5.43 (4.24, 6.61)* |
| PA | M | Low | Ram,2000 | 5 | 114 | Moderate | 5.56 (3.94, 7.19)* |
| PA | A | Low | Feng, 2021 | 3 | 141 | Moderate | 4.45 (3.32, 5.58)* |
| PA | A | Moderate | Shi, 2022 | 5 | 237 | Critically low | 1.18 (0.87, 1.48)* |
| PA | C | Very Low | Yang,2022 | 4 | 149 | Moderate | 5.86 (2.53, 9.19)* |

**Fig 6. A. The forest plot of 6MWD.** The forest plot of VO2max.

| Intervention | Ages | GRADE | Author | N | n | Quality | | MD/SMD (95% CI) |
|---|---|---|---|---|---|---|---|---|
| IMT | M | Low | Chen, 2022 | 4 | 136 | Critically low | | 3.32 (1.73, 4.91)* |
| IMT | M | Low | Lista, 2022 | 9 | 214 | Low | | 21.95 (15.05, 28.85)* |
| IMT | M | Very Low | Wang, 2022 | 7 | 202 | Critically low | | 27.62 (6.50, 48.74)* |
| IMT | M | Very Low | You,2024 | 6 | 183 | Critically low | | 26.21 (1.71, 50.72)* |
| IMT | A | Low | Ram, 2003 | 3 | 76 | High | | 23.07 (15.65, 30.50)* |
| IMT | A | Low | Silva, 2013 | 4 | 84 | High | | 13.34 (4.70, 21.98)* |
| IMT | C | Very Low | Xiang,2024 | 5 | 227 | Critically low | | 25.36 (2.47, 48.26)* |

MD/SMD(95% CI)

**Fig 7. The forest plot of PImax.**

modalities produced distinct advantages: aerobic exercise (AE) enhanced AQLQ and lung function (FEV1, FVC, PEF); breathing exercises (BE) improved AQLQ, FVC, and maximal inspiratory pressure ($PI_{max}$); yoga correlated with enhancements in AQLQ, FEV1, and FVC; aquatic exercise resulted in elevated FEV1; and inspiratory muscle training (IMT) facilitated improvements in FEV1, FVC, PEF, and PImax. However, despite these favorable results, the overall quality of the evidence continues to be inadequate.

## Assessment of the quality of the included SRs

The methodological quality of the included SRs was assessed using the AMSTAR 2 tool. Of the 34 SRs, 10 were assessed as moderate to high quality, while 24 were categorized as low or critically low quality. The essential domains with compliance below 70% comprised items 2, 7, and 15, whereas further non-essential items 3, 10, and 16 exhibited methodological shortcomings. A significant limitation noted was that 88.2% of SRs did not disclose funding sources for the primary studies included. Prior research indicates that industry-funded studies are more prone to report favorable effectiveness outcomes compared to non-industry-funded studies [46]. The lack of funding source transparency may impede decision-making by obstructing the evaluation of possible conflicts of interest. Moreover, 44.1% of SRs failed to provide explicit justifications for their inclusion criteria regarding study types, merely stating the inclusion of RCTs. The omission of studies without justification constrains transparency and may compromise the reliability of SRs [22]. Significantly, 61.7% of SRs in this study offered justifications for the exclusion of literature but failed to provide a comprehensive list of excluded studies. A detailed exclusion list with rationales is essential to assess the influence of excluded studies on the results. Furthermore, 50.0% of the included SRs lacked adequate information concerning prior protocol registration or publication. The lack of a registered protocol heightens the risk of bias and diminishes the objectivity of SRs [47]. While the majority of SRs used suitable methodologies to evaluate the risk of bias in individual studies, 35.3% neglected to consider the potential impact of bias on the interpretation of their results. Considering that bias in primary studies can profoundly influence the conclusions of SRs, assessing its impact is crucial for establishing the reliability of the evidence [48].

## Overall completeness and applicability of the evidence

The GRADE framework was employed to evaluate the certainty of evidence for 102 outcome measures. The assessment indicated that 16 outcomes (15.6%) possessed moderate certainty, 48 outcomes (47.0%) exhibited low certainty, and 38 outcomes (37.2%) demonstrated very low certainty. The principal factors for reducing evidence quality comprised study limitations, imprecision, and publication bias. An examination of the original clinical trials revealed that although randomization methods were generally documented, allocation concealment and blinding were frequently insufficiently detailed. While blinding participants in physical activity interventions poses difficulties, ensuring the blinding of outcome assessors and data analysts could enhance methodological rigor. As a result, SRs that included these studies inevitably experienced a reduction in evidence quality. Significant heterogeneity was noted among studies owing to differences in asthma

severity, exercise modalities, intensity, frequency, and duration. The lack of precision in evidence quality was mainly due to broad confidence intervals and limited sample sizes. To improve the quality of evidence, forthcoming clinical trials and SRs must conform to standardized methodologies and adhere to AMSTAR and GRADE guidelines. Enhancing methodological rigor will augment the reliability of findings, thus informing clinical practice and addressing current evidence deficiencies in asthma management.

## Implications for clinicians and research

The utilization of complementary and alternative medicine in asthma management is receiving worldwide acknowledgment. Physical activity, a scalable and economical adjunctive therapy, has been endorsed in the GINA report [1]. Physical activity encompasses any movement that expends energy, while exercise training denotes a systematic and organized type of physical activity aimed at improving physical fitness [49]. Many SRs have assessed the impact of physical activity on asthma-related outcomes, including quality of life, lung function, exercise capacity, and respiratory muscle strength. However, the results from these reviews are not wholly consistent. Huang F et al. indicated that exercise training markedly enhances the quality of life, exercise endurance, and pulmonary function in asthma patients. Nevertheless, the review omitted unstructured physical activity and failed to examine the varying effects of specific exercise modalities on health outcomes [50]. To our knowledge, this study is the inaugural systematic synthesis of evidence regarding the role of physical activity in asthma management, while concurrently addressing asthma severity, patient age, and specific exercise modalities. Aerobic exercise, yoga, breathing training, and respiratory muscle training have garnered the most focus among the examined exercise modalities [14,28,31,37]. Evidence suggests that diverse forms of physical activity produce unique impacts on different health outcomes. Most SRs employ the AQLQ to evaluate patients' quality of life, and nearly all indicate enhancements subsequent to physical activity interventions, with the exception of those that concentrate solely on respiratory muscle training. Conversely, inspiratory muscle training seems especially efficacious in enhancing objective lung function metrics. SRs investigating breathing exercises and yoga typically do not demonstrate substantial enhancements in FEV1 and FVC, with some even suggesting negative trends. Consequently, to attain holistic enhancements in both subjective and objective results, the formulation of a multifaceted physical activity program is advised. In addition, AQLQ outcomes are derived from self-reported data, whereas lung function measures are based on objective physiological assessments. While enhancements in AQLQ are advantageous, dependence on self-reported outcomes may introduce bias and potentially exaggerate the actual impact of physical activity on health. Consequently, forthcoming research should utilize a blend of subjective and objective outcome measures to augment the reliability and validity of their results. Asthma often results in diminished physical activity levels [51], but limited research has investigated methods to improve adherence to physical activity in asthma patients. Based on this observation, we suggest that subsequent research investigate the impact of various physical activity programs on patient adherence, which may guide the creation of more sustainable and user-friendly exercise regimens. Physical activity has demonstrated beneficial effects on quality of life, pulmonary function, exercise capacity, respiratory muscle strength, and other dimensions. However, a considerable portion of the evidence is of low or very low quality. In light of these concerns, more rigorous, well-structured, large-scale randomized controlled trials are essential to yield superior evidence and to further evaluate the efficacy of physical activity in asthma management. Firstly, original RCTs must disclose randomization techniques, allocation concealment, and blinding, in accordance with CONSORT (Consolidated Standards of Reporting Trials) [52], enabling clinical practitioners to assess the evidence with greater precision. Secondly, the specific frequency, intensity, duration, and type of activity to be undertaken based on asthma severity, subtype, or diverse populations of asthma patients are essential factors that must be established. Thirdly, all SRs must be registered in advance to improve transparency in the process and mitigate the risk of methodological bias. In addition, significant variability is present among studies regarding asthma severity, exercise modality, intensity, frequency, and intervention duration. This variability hinders the capacity to establish conclusive determinations about the ideal type and dosage of physical activity for asthma management. Consequently, further high-quality

primary studies that meticulously account for these variables are necessary to formulate definitive, evidence-based recommendations for specific asthma populations.

### Study limitations

Several constraints must be acknowledged when analyzing the results of this study. First, the methodological quality of the included SRs was predominantly inadequate, which diminished the overall certainty of the evidence. Second, although this review performed subgroup analyses by exercise interventions and offered a descriptive synthesis of age, significant heterogeneity was present among studies concerning asthma severity, exercise modalities, intensity, frequency, and intervention duration. This heterogeneity constrained the capacity to formulate conclusive determinations concerning the ideal exercise protocol (type and dosage) for asthma management.

### Conclusion

The overview showed that physical activity may improve asthma-related outcomes, encompassing quality of life (AQLQ), asthma control, pulmonary function ($FEV_1$, FVC, PEF), exercise capacity ($VO_2max$, 6MWD), and PImax. Specific interventions including inspiratory muscle training (IMT), yoga, breathing exercises, and aquatic exercises, exhibit focused enhancements in specific areas. Since most clinical evidence was low quality, there remains an urgent need for rigorous studies to strengthen the evidence base. Additionally, acknowledging the individual variability among asthma patients, formulating customized exercise prescriptions for each patient would be clinically advantageous.

### Supporting information

**S1 Appendix. Search Strategy.**
(DOCX)

**S2 Appendix. Studies identified in the literature search.**
(DOCX)

**S3 Appendix. Evidence quality of GRADE system.**
(DOCX)

**S4 Data. Data extracted from included studies and used for all analyses.**
(XLSX)

**S1 File. PRISMA_2020_checklist.**
(DOC)

### Author contributions

**Conceptualization:** Wenrui Liu.

**Data curation:** Wenrui Liu, Zhenzhen Feng, Shangyue Song.

**Formal analysis:** Wenrui Liu.

**Funding acquisition:** Zhenzhen Feng.

**Methodology:** Wenrui Liu, Zhenzhen Feng, Shangyue Song, Siyuan Lei.

**Software:** Wenrui Liu.

**Supervision:** Zhenzhen Feng, Siyuan Lei, Wen Guo.

**Visualization:** Zhenzhen Feng.

**Writing – original draft:** Wenrui Liu.

**Writing – review & editing:** Wenrui Liu.

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
