## [Decision Letter · Decision Letter 0]

Dear Dr. Feng,

Thank you for submitting your manuscript to PLOS ONE. After careful consideration, we feel that it has merit but does not fully meet PLOS ONE’s publication criteria as it currently stands. Therefore, we invite you to submit a revised version of the manuscript that addresses the points raised during the review process.

We look forward to receiving your revised manuscript.

Kind regards,

Hidetaka Hamasaki

Academic Editor

PLOS ONE

Journal Requirements:

4. As required by our policy on Data Availability, please ensure your manuscript or supplementary information includes the following: 

Reviewers' comments:

Reviewer's Responses to Questions

**Comments to the Author**

1. Is the manuscript technically sound, and do the data support the conclusions?

Reviewer #1: Partly

Reviewer #2: No

Reviewer #3: Yes

2. Has the statistical analysis been performed appropriately and rigorously?

Reviewer #1: N/A

Reviewer #2: Yes

Reviewer #3: Yes

3. Have the authors made all data underlying the findings in their manuscript fully available?

Reviewer #1: Yes

Reviewer #2: Yes

Reviewer #3: Yes

4. Is the manuscript presented in an intelligible fashion and written in standard English?

Reviewer #1: No

Reviewer #2: Yes

Reviewer #3: Yes

Reviewer #1: Overall comments

Overall, the language used is unclear and often incomprehensible, with numerous grammatical errors and inconsistent formatting e.g. line and between-word spacing.

Specific comments

Title

• Make it more clearly if you mean asthma treatment or prevention or what perspectives?

Abstract

• What is the targeted population?

Intro section

• P3 L54-58 this sentence is too long to follow. Try breaking down.

• P3-L58-60 this sentence is incomprehensible

• P3-L60 “An” overview, not “a”

Method section

• Identify the keywords used to search for papers.

• Any limitation about the language of publication included in the review

• Identify age of the samples

• Identify clearly if the samples were people with current asthma condition or patients at hospital or?

• Identify clearly study design that were included in the review.

• Identify clearly measurement of PA whether the study included both device-based and subjective approaches.

• This study used only 2 reviewers. Usually 3 reviewers are used to screen the papers, in case there are some conflicting ideas when trying to classify the papers. Explain strategies in managing the challenges with 2 reviewers.

Result section

• Identify the colours use in figure 3 and what they represented

• Figure 2 showed that there were so many critically low-quality papers, why were they included in the study?

Conclusions

• The conclusion didn’t make a good sense of what have been found from the review.

Reviewer #2: This was an interesting review that evaluates the effect of physical activity on quality of life, asthma control, lung function, exercise performance and muscle strength in people with asthma. The manuscript is certainly unique in its own right, however, there are some major concerns worth raising.

My primary concern with this review involves the Methods section.

The use of additional electronic academic databases would likely have added to the study in both complexity and sample size. If the scope of this study were expanded to use additional databases, more sources might have been identified and explored. For example, the following electronic databases could have been used for a more thorough and inclusive search;

“ArticleFirst; Biomed Central; BioOne; BIOSIS; CINAHL; EBSCOHost; JSTOR; ProQuest; SAGE Reference Online; Scopus; google scholar; ScienceDirect; SpringerLink; Taylor & Francis; and Wiley Online.” These databases would have likely added to the overall literature search in their academic rigor, aim, and biomedical scope. While PubMed, Embase, the Cochrane Library, and Web of Science are excellent databases, the use of more databases would have added to the study sample size.

Overall, this paper does not merit publication in its current form without a more thorough search of more databases.

Reviewer #3: Review of the Abstract:

1. While the abstract mentions that six systematic reviews were rated as moderate to high quality, it lacks specific details about the findings or overall effect sizes, which could provide more context and depth to the results. Enhance the details by including specific findings or statistics from the systematic reviews regarding the effectiveness of physical activity on measured outcomes.

2. The abstract briefly touches on limitations concerning study protocols and exclusion criteria but could benefit from a more detailed discussion of how these limitations may affect overall findings and interpretations or influence future research and clinical practice.

3. The conclusion reiterates the impact of physical activity but seems weak to summarize the implications for practice or suggest how these findings could be applied in clinical settings. Revise the conclusion to emphasize the importance of physical activity in asthma management, potentially highlighting areas for future research or clinical application.

4. Consider clarifying Methodology: Although the abstract mentions the use of AMSTAR 2 and GRADE, it could briefly explain how these tools were applied, providing clarity on the assessment process.

5. recommend concluding with a statement on the practical implications of the findings for healthcare practitioners or policymakers. This would enhance the relevance of the research and may include recommendations for physical activity guidelines for asthma patients.

Addressing these points, the abstract could become more informative and impactful, offering a clearer understanding of the study's significance and applications in the field of asthma management.

Review of Manuscript Draft:

Strengths:

1.The topic effectively addresses a significant public health issue, considering the prevalence of asthma and the limitations of current pharmacological treatments. This is crucial for both clinicians and patients.

2. The manuscript employs an overview of systematic reviews, a robust methodology for synthesizing existing evidence regarding the effectiveness of physical activity in asthma management.

3. The abstract articulates the need for a synthesis of existing systematic reviews, highlighting a notable gap in the current literature. (Clear Objectives)

4. The adherence to established guidelines (PRISMA, GRADE, AMSTAR 2) for conducting and reporting systematic reviews enhances the credibility of the research.

Weaknesses:

1. Manuscript Length: The abstract is overly lengthy and could be more concise. Key findings and implications should be summarized more effectively.

2. Lack of Specific Results: Although the methodology is well-articulated, the manuscript seems weak to present specific results or outcomes from the systematic reviews, which are essential for understanding the effectiveness of physical activity in asthma.

3. Clarity of Structure: The manuscript would benefit from clearer segmentation among sections (introduction, methods, results, and conclusion) to improve overall readability.

4. Language and Grammar: Several grammatical errors and awkward phrases need refinement for clarity (e.g., “but the concern is the side effects of long-term use of ICS” could be expressed more clearly).

Areas for Improvement & Recommendations:

1. Revise Manuscript for Conciseness: Rework the manuscript to focus on critical elements, ensuring it is succinct and directly addresses the research questions and findings. Create a structured abstract with distinct sections (Background, Methods, Results, Conclusions), clearly outlining each component.

2. Inclusion of Results: Incorporate specific findings from the systematic reviews to provide a snapshot of the evidence regarding the effectiveness of physical activity on asthma outcomes. Include key statistics or findings to substantiate the effectiveness of physical activity in asthma management.

3. Highlighting Limitations: Briefly mention the limitations of the current evidence to provide a balanced view of the findings.

4. Future Research Directions: Consider mentioning specific areas where future research is needed or what types of studies would enhance understanding in this field.

5. Language Refinement & Proofreading: Consider thorough proofreading to enhance clarity and address grammatical issues, improving sentence structure for professionalism.

Conclusion:

Overall, the manuscript presents a relevant and potentially impactful overview of the role of physical activity in asthma management. However, improvements in conciseness, clarity, and the inclusion of specific results will enhance the quality and effectiveness of the abstract, making it a more valuable resource for readers.

**Do you want your identity to be public for this peer review?** For information about this choice, including consent withdrawal, please see our Privacy Policy

Reviewer #1: No

Reviewer #2: No

Reviewer #3: No

---

## [Author Response · Author response to Decision Letter 1]

26 Mar 2025

Response to Reviewer’s Comments

Dear Editors and Reviewers

On behalf of all the contributing authors, I would like to express our sincere appreciation of your letter and reviewers’ constructive comments concerning our article entitled “The effectiveness of physical activity in asthma: an overview of systematic review”(Manuscript ID PONE-D-24-54330). Those comments are all valuable and very helpful for revising and improving our paper, as well as the important guiding significance to our researches. In the following, we have provided detailed responses to each of the reviewers' comments. Revised portion are marked in blue in the paper. Additionally, all relevant data are within the paper and its Supporting Information files. In this response letter, the reviewers' comments are presented in italics, and our corresponding changes and additions to the manuscript are highlighted in blue text. If we missed any one of the comments, please let us know. This document includes our responses to your comments, and the revised portion are presented in our main document. We have tried our best to make all the revisions clear, and we hope that the revised manuscript meets the requirements for publication.

Thanks again for your consideration of publishing our manuscript in your journal. We are looking forward to hearing from you soon.

Best wishes,

Sincerely,

Zhenzhen Feng

Contents

Reviewer #1 4

Reviewer #2 8

Reviewer #3 10

Reviewer #1

Comment 1: Overall, the language used is unclear and often incomprehensible, with numerous grammatical errors and inconsistent formatting e.g. line and between-word spacing.

Response 1: Thanks for the suggestion. We have collaborated with a certified professional editing service to conduct rigorous proofreading and substantive revisions during this round of manuscript preparation. The text has been systematically refined to ensure grammatical accuracy, improve readability, and align with academic writing standards. A copy of the language-editing certificate issued by the service provider has been uploaded to the Supplementary Materials section.

Comment 2: Title: Make it more clearly if you mean asthma treatment or prevention or what perspectives?

Response 2: Thank you for your valuable feedback. We appreciate your suggestion to clarify the scope of our review. As recommended, we have revised the title to "The effectiveness of physical activity in asthma management: An overview of systematic reviews" to better reflect the comprehensive nature of our analysis, which examines multiple perspectives including asthma quality of life, asthma control, lung function, exercise capacity, and respiratory muscle strength. Should you feel further refinements would enhance clarity, we would be grateful for any additional guidance. We believe this modified title more accurately represents the synthesized evidence from systematic reviews regarding PA's multidimensional role in asthma care.

Comment 3: What is the targeted population?

Response 3: Thank you for raising this important point. We have refined our inclusion criteria to explicitly state: "Participants diagnosed with asthma, irrespective of age, sex, or severity."(Page4, Line 98-99).

Comment 4:(1) P3 L54-58 this sentence is too long to follow. Try breaking down. (2) P3-L58-60 this sentence is incomprehensible (3) P3-L60 “An” overview, not “a”.

Response 4: Thank you for your meticulous review.We have checked and revised this paragraph and it will read as follows Physical activity, a non-pharmacological intervention, has been widely used to improve quality of life, mitigate asthma symptoms, and enhance exercise capacity, cardiopulmonary fitness, and muscular strength. The World Health Organization characterizes physical activity as "any bodily movement generated by skeletal muscles that necessitates energy expenditure exceeding resting levels." Numerous SRs have investigated the effects of physical activity on asthma. However, their results are not wholly consistent and have not been thoroughly synthesized . Consequently, a synthesis derived from published SRs is crucial to mitigate the shortcomings of individual studies and offer a more thorough comprehension of the impacts of physical activity on asthma.(Page3, Line 64-73).

Comment 5: Method section: (1) Identify the keywords used to search for papers. (2)Any limitation about the language of publication included in the review.

Response 5: Thank you for your suggestion, and we have added a description of the language limitations, search strategy, age of the study population, and the type of original study design that you mentioned, and the revised content is as follows:No restrictions were placed on the origin of the SRs, the date of publication, or the language of publication. The search strategy included relevant words related to asthma and physical activity (including 'physical activity', 'exercise training', 'exercise therapy', 'endurance training', 'resistance training', 'muscle stretching exercises', 'upper limb training', 'Running', 'Jogging', 'Swimming', 'Walking', 'Qigong', 'Yoga', 'Pilates' 'Gymnastics', 'Neuromuscular Electrical Stimulation', 'Physiotherapy') and Systematic Review and was adjusted for each database. Comprehensive search methodologies are outlined in Appendix S1.�Page3-4, Line 85-92

Comment 6: Method section: (1) Identify age of the samples (2) Identify clearly if the samples were people with current asthma condition or patients at hospital or? (3) Identify clearly study design that were included in the review. (4) Identify clearly measurement of PA whether the study included both device-based and subjective approaches.

Response 6: Thank you for your suggestion, we have added information such as the age of the study population you mentioned, the type of original study design, etc., to the inclusion criteria, and the revised content is as follows:The inclusion criteria encompassed all the subsequent aspects: (1) Participants diagnosed with asthma, irrespective of age, sex, or severity; (2) the intervention encompassed any form of physical activity(defined as any bodily movement generated by skeletal muscles that necessitates energy expenditure exceeding resting levels); (3) studies compared physical activity against placebo, blank control, usual care, conventional drug therapy, or other non-pharmacological therapies; (4) at least one outcome was measured among quality of life (AQLQ), asthma control (ACQ), forced expiratory volume in one second (FEV1), forced vital capacity (FVC), peak expiratory flow (PEF), six-minute walk distance (6MWD), maximal oxygen consumption (VO2max), and maximal inspiratory pressure (PImax); (5) The SRs were founded on randomized controlled trials concerning physical activity for asthma. (P4, Line98-108)

Comment 7: This study used only 2 reviewers. Usually 3 reviewers are used to screen the papers, in case there are some conflicting ideas when trying to classify the papers. Explain strategies in managing the challenges with 2 reviewers.

Response 7�Thank you for raising this important point. We sincerely appreciate your attention to methodological rigor. To clarify, our manuscript explicitly outlines the screening process in the Study Selection section:The eligibility screening was performed independently by two researchers, Wenrui Liu and Shangyue Song. A third author, Zhenzhen Feng, was designated to resolve any disputes concerning screening. (Page 4, Lines 94–97)

Comment 8: Result section:Identify the colours use in figure 3 and what they represented.

Response 8: We sincerely apologize for the confusion caused by the unclear color annotations in the original Figure 3. To enhance clarity and conciseness, we have removed the bubble plot and the GRADE evaluation table. Instead, we now present the results through forest plots (revised Figure 3-7), which clearly display effect sizes, 95% confidence intervals, statistical significance, and GRADE assessment outcomes. Detailed GRADE evaluation procedures are provided in Supplementary File S3 to ensure transparency. These revisions aim to improve the clarity of the study results. We deeply appreciate your constructive feedback and welcome any further suggestions to refine this work.

Comment 9 Figure 2 showed that there were so many critically low-quality papers, why were they included in the study?

Response 9: Thank you for your thoughtful observation. We acknowledge that a significant proportion of included studies were rated as "critically low-quality" in the methodological assessment (Figure 2). This reflects a broader challenge in the field of physical activity research, where methodological rigor remains inconsistent—a key finding highlighted in our discussion. Excluding low-quality studies prematurely could have skewed the representation of existing evidence and obscured critical gaps in the literature. Instead, we analyzed these studies’ limitations in the discussion section (Page 18, Lines 335–342) to contextualize their impact on overall conclusions and advocate for improved methodological standards in future research. We appreciate your feedback, as it underscores the importance of addressing quality heterogeneity in evidence synthesis. Please let us know if further clarification is needed.

Comment 10: Conclusions: The conclusion didn’t make a good sense of what have been found from the review.

Response 10: Thanks for your careful suggestion, we have revised the conclusion section. Physical activity may be an effective complementary therapy for asthma. IMT, yoga, breathing exercises, and aquatic exercises, which show focused enhancement in specific areas. Since most clinical evidence was low quality, there remains an urgent need for rigorous studies to strengthen the evidence base.(P2, L41-44).

Reviewer #2

Comment 1: This was an interesting review that evaluates the effect of physical activity on quality of life, asthma control, lung function, exercise performance and muscle strength in people with asthma. The manuscript is certainly unique in its own right, however, there are some major concerns worth raising.My primary concern with this review involves the Methods section.The use of additional electronic academic databases would likely have added to the study in both complexity and sample size. If the scope of this study were expanded to use additional databases, more sources might have been identified and explored. For example, the following electronic databases could have been used for a more thorough and inclusive search; “ArticleFirst; Biomed Central; BioOne; BIOSIS; CINAHL; EBSCOHost; JSTOR; ProQuest; SAGE Reference Online; Scopus; google scholar; ScienceDirect; SpringerLink; Taylor & Francis; and Wiley Online.” These databases would have likely added to the overall literature search in their academic rigor, aim, and biomedical scope. While PubMed, Embase, the Cochrane Library, and Web of Science are excellent databases, the use of more databases would have added to the study sample size. Overall, this paper does not merit publication in its current form without a more thorough search of more databases.

Response 1: Thank you for your thoughtful feedback and guidance on strengthening our literature search. To ensure a more comprehensive and interdisciplinary review, we added 8 specialized databases beyond our initial search (PubMed, Embase, Cochrane Library, Web of Science): PEDro (focused on physical therapy and exercise intervention studies), CINAHL (nursing and allied health literature, critical for patient-reported outcomes), Scopus (multidisciplinary coverage, including non-English and grey literature), Sportdiscus (sport science research for exercise prescription insights), Chinese databases (CNKI, Wan Fang, SinoMed, VIP) to capture China-specific evidence. The date of the search was updated to 2025.2.1. In addition, during this supplementary search, we discovered some valuable literature regarding the efficacy of physical activity on childhood asthma. In previous studies, due to limited searches, we found that the quality of articles on childhood asthma was generally low, and considering that we only conducted a descriptive summary, the heterogeneity of childhood asthma was quite high, thus we excluded studies that solely focused on childhood asthma. This search and selection expanded the age range targeted by the review, ultimately adding 17 articles. Among them, there were 9 articles related to adults and mixed populations, and 8 articles specifically concerning childhood asthma. The screening process for supplementary retrieval is shown in Figure 1 (P6), and the list of excluded literature can be found in Appendix S2.

Thank you again for your affirmation and support of our work and manuscript, and thank you for your contribution and dedication to our manuscript. Your affirmation and encouragement are the source of our unremitting progress. As a researcher, it is our common pursuit to produce more researches that are of interest to readers and have clinical value. With your help and guidance, the quality of our manuscripts has been qualitatively improved, and we are looking forward to your further suggestions and criticisms.

Reviewer #3

Comment 1: Revise Manuscript for Conciseness: Rework the manuscript to focus on critical elements, ensuring it is succinct and directly addresses the research questions and findings. Create a structured abstract with distinct sections (Background, Methods, Results, Conclusions), clearly outlining each component.

Response 1: Thank you for your valuable guidance on improving manuscript clarity. We have restructured the abstract to align with your recommendations, adopting a formalized framework with distinct sections: Background: Physical activity (PA) has become a promising complementary non-pharmacological intervention to improve exercise capacity, cardiopulmonary fitness, and quality of life in individuals with asthma. This overview systematically consolidates existing evidence to assess the clinical viability of physical activity as a scalable supplementary therapy for asthma management. Methods: We searched 12 electronic databases to identify systematic reviews (SRs) from inception until February 12, 2025, concerning the efficacy of physical activity in asthma management. Literature was independently reviewed, data extracted and verified by two researchers. A third author was designated to mediate any disputes concerning screening. The methodological quality of SRs was assessed using the A Measurement Tool to Assess SRs 2 (AMSTAR 2) checklist, and the certainty of evidence for key outcomes was evaluated using the Grading of Recommendations Assessment, Development and Evaluation (GRADE) framework.

Results: This study analyzed 34 SRs (published 2000–2024) that included quantitative synthesis involving 113–2280 participants from asthma populations: adults (9 SRs), children (8 SRs), and mixed adult-child cohorts (17 SRs), with disease severity varying from mild to severe. Ten SRs were assessed as moderate to high quality by AMSTAR 2, whereas the other SRs were classified as low or very low quality. We evaluated the quality of evidence for SRs utilizing the GRADE evidence quality assessment framework. Thirteen moderate-quality evidence, and 51 low or very low-quality evidence support the improvement of PA on the outcomes of asthma quality of life, asthma control, lung function, exercise capacity, and respiratory muscle strength. Conclusion: Physical activity may be an effective complementary therapy for asthma. IMT, yoga, breathing exercises, and aquatic exercises, which show focused enhancement in specific areas. Since most clinical evidence was low quality, there remains an urgent need for rigorous studies to strengthen the evidence base.(P1-2, L14-44)

Comment 2:While the abstract mentions that six systematic reviews were rated as moderate to high quality, it lacks specific details about the findings or overall effect sizes, which could provide more context and depth to the results. Enhance the details by including specific findings or statistics from the systematic reviews regarding the effectiveness of physical activity on measured outcomes.

Response 2 Thank you for your valuable suggestion. To address the need for

---

## [Decision Letter · Decision Letter 1]

The effectiveness of physical activity in asthma management: An overview of systematic reviews

Dear Dr. Feng,

Thank you for submitting your manuscript to PLOS ONE. After careful consideration, we feel that it has merit but does not fully meet PLOS ONE’s publication criteria as it currently stands. Therefore, we invite you to submit a revised version of the manuscript that addresses the points raised during the review process.

We look forward to receiving your revised manuscript.

Kind regards,

Hidetaka Hamasaki

Academic Editor

PLOS ONE

Journal Requirements:

Additional Editor Comments:

Reviewer #2: thank you for addressing these comments. the paper is much improved.

Reviewer #3: The document presents a comprehensive overview of the effectiveness of physical activity in asthma management, and your request for feedback on the conclusion, methodology, and findings is well noted.

Methodology: The methodology is generally well-structured, detailing the search strategy and the criteria for including systematic reviews. However, there are areas for improvement. For instance, while the document mentions the use of AMSTAR 2 and GRADE frameworks for assessing quality, it could provide more clarity on how these assessments influenced the selection of studies. Furthermore, the inclusion of a broader range of databases, as suggested by reviewers, could enhance the comprehensiveness of the literature search. Addressing the limitations regarding language and publication bias in more detail would also strengthen the methodology section.

Findings: The findings are presented clearly, with a good overview of the number of systematic reviews analyzed and the participant demographics. However, the results could be enhanced by including more specific data points, such as effect sizes and confidence intervals, to provide a clearer picture of the impact of physical activity on asthma outcomes. Additionally, discussing the implications of the findings in relation to existing literature would provide context and highlight the significance of the results. Overall, the document is a valuable contribution to the field, but refining these sections could improve clarity and impact.

The conclusion effectively summarizes the potential of physical activity as a complementary therapy for asthma. However, it could benefit from a more explicit articulation of the implications of the findings. For instance, it might be helpful to emphasize the specific types of physical activities that showed the most promise and how they can be integrated into standard asthma management practices. Additionally, reiterating the need for further research could be strengthened by suggesting specific areas where more rigorous studies are necessary.

---

## [Author Response · Author response to Decision Letter 2]

24 Apr 2025

Reviewer #2

Comment 1: Thank you for addressing these comments. the paper is much improved.

Response 1: We sincerely appreciate your thoughtful feedback. We are genuinely appreciative of the time and expertise you dedicated to evaluating our manuscript. The revisions have enhanced the paper's quality, which is encouraging to note. If any further minor adjustments are necessary, we will gladly execute them promptly.

Reviewer #3

Comment 1: The document presents a comprehensive overview of the effectiveness of physical activity in asthma management, and your request for feedback on the conclusion, methodology, and findings is well noted. Methodology: The methodology is generally well-structured, detailing the search strategy and the criteria for including systematic reviews. However, there are areas for improvement. For instance, while the document mentions the use of AMSTAR 2 and GRADE frameworks for assessing quality, it could provide more clarity on how these assessments influenced the selection of studies.

Response 1: We value your meticulous review and insightful feedback concerning the methodological rigour of our study. We appreciate the opportunity to elucidate this significant matter. Targeted reaction to methodology feedback are as follow: You accurately noted that, although we utilized the AMSTAR 2 and GRADE frameworks to evaluate the methodological quality and certainty of evidence in the included systematic reviews, these evaluations were not applied as exclusion criteria. This methodology was intentional and predetermined in our PROSPERO protocol (CRD42024520761), wherein the inclusion criteria were found on study design attributes (systematic reviews/meta-analyses) and their pertinence to asthma and physical activity, rather than quality benchmarks. Our rationale was twofold: (1) to transparently present the complete range of available evidence, including lower-quality studies—with relevant caveats indicated in Fig 2 and Table S3; and (2) to prevent potential selection bias that could result from excluding studies based on post hoc quality evaluations. To improve clarity, we have incorporated the following sentence in the Methods section: The AMSTAR-2 and GRADE tools were utilized to evaluate methodological quality and the certainty of evidence in the systematic reviews included. However, these instruments were employed exclusively for assessment purposes and did not affect the criteria for study selection (Page5, Line 128-131).Thank you again for your insightful critique that helps strengthen the paper's transparency.

Comment 2:Furthermore, the inclusion of a broader range of databases, as suggested by reviewers, could enhance the comprehensiveness of the literature search. Addressing the limitations regarding language and publication bias in more detail would also strengthen the methodology section.

Response 2 We express our sincere gratitude to the reviewer for this astute suggestion, which has enhanced the clarity of our methodology. We concur that explicit discourse on potential limitations fortifies the rigour of systematic reviews. Concerning issues of language and publication limitations, we clearly articulated in the Methods section: No restrictions were placed on the origin of the SRs, the date of publication, or the language of publication.(Page4, Line 91-92). We are especially grateful for the opportunity to refine our methodological rigor through this invaluable feedback.

Comment 3:The findings are presented clearly, with a good overview of the number of systematic reviews analyzed and the participant demographics. However, the results could be enhanced by including more specific data points, such as effect sizes and confidence intervals, to provide a clearer picture of the impact of physical activity on asthma outcomes.

Response 3: We appreciate your valuable recommendation concerning the presentation of results. Consequently, we have amended the Results section to incorporate forest plots that explicitly present essential information from the systematic reviews included. The visualizations encompass the quantity of studies examined in each systematic review, sample sizes, effect sizes (MD/SMD) with 95% confidence intervals, and GRADE assessments of evidence quality (Pages 12-17, Figures 3-7).

Comment 4: Additionally, discussing the implications of the findings in relation to existing literature would provide context and highlight the significance of the results. Overall, the document is a valuable contribution to the field, but refining these sections could improve clarity and impact.

Response 4�We value your constructive feedback concerning the Discussion section. This section has been augmented to offer a more profound analysis of our findings within the framework of existing literature. Additionally, we have amended the References section to accurately represent the revised citations: The utilization of complementary and alternative medicine in asthma management is receiving worldwide acknowledgment. Physical activity, a scalable and economical adjunctive therapy, has been endorsed in the GINA report [1]. Physical activity encompasses any movement that expends energy, while exercise training denotes a systematic and organized type of physical activity aimed at improving physical fitness[50]. Many SRs have assessed the impact of physical activity on asthma-related outcomes, including quality of life, lung function, exercise capacity, and respiratory muscle strength. However, the results from these reviews are not wholly consistent. Huang F et al. indicated that exercise training markedly enhances the quality of life, exercise endurance, and pulmonary function in asthma patients. Nevertheless, the review omitted unstructured physical activity and failed to examine the varying effects of specific exercise modalities on health outcomes[51]. To our knowledge, this study is the inaugural systematic synthesis of evidence regarding the role of physical activity in asthma management, while concurrently addressing asthma severity, patient age, and specific exercise modalities. Aerobic exercise, yoga, breathing training, and respiratory muscle training have garnered the most focus among the examined exercise modalities[14, 29, 32, 38]. Evidence suggests that diverse forms of physical activity produce unique impacts on different health outcomes. Most SRs employ the AQLQ to evaluate patients' quality of life, and nearly all indicate enhancements subsequent to physical activity interventions, with the exception of those that concentrate solely on respiratory muscle training. Conversely, inspiratory muscle training seems especially efficacious in enhancing objective lung function metrics. SRs investigating breathing exercises and yoga typically do not demonstrate substantial enhancements in FEV1 and FVC, with some even suggesting negative trends. Consequently, to attain holistic enhancements in both subjective and objective results, the formulation of a multifaceted physical activity program is advised. In addition, AQLQ outcomes are derived from self-reported data, whereas lung function measures are based on objective physiological assessments. While enhancements in AQLQ are advantageous, dependence on self-reported outcomes may introduce bias and potentially exaggerate the actual impact of physical activity on health. Consequently, forthcoming research should utilize a blend of subjective and objective outcome measures to augment the reliability and validity of their results. Asthma often results in diminished physical activity levels [52], but limited research has investigated methods to improve adherence to physical activity in asthma patients. Based on this observation, we suggest that subsequent research investigate the impact of various physical activity programs on patient adherence, which may guide the creation of more sustainable and user-friendly exercise regimens. Physical activity has demonstrated beneficial effects on quality of life, pulmonary function, exercise capacity, respiratory muscle strength, and other dimensions. However, a considerable portion of the evidence is of low or very low quality. In light of these concerns, more rigorous, well-structured, large-scale randomized controlled trials are essential to yield superior evidence and to further evaluate the efficacy of physical activity in asthma management. Firstly, original RCTs must disclose randomization techniques, allocation concealment, and blinding, in accordance with CONSORT (Consolidated Standards of Reporting Trials) [53], enabling clinical practitioners to assess the evidence with greater precision. Secondly, the specific frequency, intensity, duration, and type of activity to be undertaken based on asthma severity, subtype, or diverse populations of asthma patients are essential factors that must be established. Thirdly, all SRs must be registered in advance to improve transparency in the process and mitigate the risk of methodological bias. In addition, significant variability is present among studies regarding asthma severity, exercise modality, intensity, frequency, and intervention duration. This variability hinders the capacity to establish conclusive determinations about the ideal type and dosage of physical activity for asthma management. Consequently, further high-quality primary studies that meticulously account for these variables are necessary to formulate definitive, evidence-based recommendations for specific asthma populations (Page19-20, Line 331-384).

Comment 5: The conclusion effectively summarizes the potential of physical activity as a complementary therapy for asthma. However, it could benefit from a more explicit articulation of the implications of the findings. For instance, it might be helpful to emphasize the specific types of physical activities that showed the most promise and how they can be integrated into standard asthma management practices. Additionally, reiterating the need for further research could be strengthened by suggesting specific areas where more rigorous studies are necessary.

Response 5�Thank you for your perceptive feedback on the Conclusion. We genuinely value the time and effort you dedicated to improving the clarity and effectiveness of our manuscript. In accordance with your recommendations, we have amended the Conclusion section as follows: Engagement in physical activity has been demonstrated to markedly enhance asthma-related outcomes. Specific interventions provide targeted advantages in various domains: aerobic exercise enhances AQLQ scores and lung function metrics, including FEV1, FVC, PEF; breathing exercises improve AQLQ, FVC, and PImax; yoga correlates with enhancements in AQLQ, FEV1, and FVC; aquatic exercise results in increased FEV1; and inspiratory muscle training yields improvements in FEV1, FVC, PEF, and PImax. Nonetheless, there exists an imperative requirement for more stringent studies to fortify the existing evidence base. Furthermore, due to the significant individual variability among asthma patients, the creation of personalized exercise prescriptions is expected to produce enhanced clinical results (Page 2, Line 41-50).

---

## [Decision Letter · Decision Letter 2]

The effectiveness of physical activity in asthma management: An overview of systematic reviews

PONE-D-24-54330R2

Dear Dr. Feng,

We’re pleased to inform you that your manuscript has been judged scientifically suitable for publication and will be formally accepted for publication once it meets all outstanding technical requirements.

Kind regards,

Hidetaka Hamasaki

Academic Editor

PLOS ONE
---

## [Editor Report · Acceptance letter]

PONE-D-24-54330R2

PLOS ONE

Dear Dr. Feng,

I'm pleased to inform you that your manuscript has been deemed suitable for publication in PLOS ONE. Congratulations! Your manuscript is now being handed over to our production team.

Kind regards,

on behalf of

Dr. Hidetaka Hamasaki

Academic Editor

PLOS ONE